# A large scale randomized controlled trial on herding in peer-review discussions

**Ivan Stelmakh**[1,2]*, **Charvi Rastogi**[3], **Nihar B. Shah**[3], **Aarti Singh**[3], **Hal Daumé III**[4,5]

**1** New Economic School, Moscow, Russia, **2** Yakov & Partners, Moscow, Russia, **3** Machine Learning Department, Carnegie Mellon University, Pittsburgh, Pennsylvania, United States of America, **4** Department of Computer Science, University of Maryland, College Park, Maryland, United States of America, **5** Microsoft Research, New York City, New York, United States of America

\* istelmakh@nes.ru

**Data Availability Statement:** In our work, we analyze data obtained from the scientific conference ICML (International conference on Machine learning) for publication of research in ML. For this, we worked in collaboration with the

## Abstract

Peer review is the backbone of academia and humans constitute a cornerstone of this process, being responsible for reviewing submissions and making the final acceptance/rejection decisions. Given that human decision-making is known to be susceptible to various cognitive biases, it is important to understand which (if any) biases are present in the peer-review process, and design the pipeline such that the impact of these biases is minimized. In this work, we focus on the dynamics of discussions between reviewers and investigate the presence of herding behaviour therein. Specifically, we aim to understand whether reviewers and discussion chairs get disproportionately influenced by the first argument presented in the discussion when (in case of reviewers) they form an independent opinion about the paper before discussing it with others. In conjunction with the review process of a large, top tier machine learning conference, we design and execute a randomized controlled trial that involves 1,544 papers and 2,797 reviewers with the goal of testing for the conditional causal effect of the discussion initiator's opinion on the outcome of a paper. Our experiment reveals no evidence of herding in peer-review discussions. This observation is in contrast with past work that has documented an undue influence of the first piece of information on the final decision (e.g., anchoring effect) and analyzed herding behaviour in other applications (e.g., financial markets). Regarding policy implications, the absence of the herding effect suggests that the current status quo of the absence of a unified policy towards discussion initiation does not result in an increased arbitrariness of the resulting decisions.

## 1 Introduction

Peer review is the backbone of academia. As a means of pre-publication verification of scientific works, it has been widely adopted by conferences and journals across many fields of science. In addition to being the cornerstone for the dissemination of completed research, peer review nowadays plays a crucial role in shaping the directions of future research: it is used by funding bodies around the world (including US agencies NSF and NIH, and European Research Council) to distribute multi-billion dollar budgets through grants and awards. Given the considerable impact the peer-review process has on the progression of science, it is crucial

conference organizers of ICML 2020 (IvanStelmakh, Hal Daumé III, Aarti Singh). We are not able to share this data in any form, because of the following reasons: 1. This is highly sensitive data concerning authors and reviewers both, where authors and reviewers provide their honest opinion with the belief that this would not affect them personally. 2. ICML 2020 is a double-blind conference and maintaining the double-blindness is an important aspect of the reviewing process. Releasing de-identified data would dilute the double-blindness. 3. It would violate the confidentiality agreement between the researchers who participated in the conference and the conference organizers. We are only able to share the aggregated statistics, which are provided in the manuscript. Further, any data requests may be sent to The International Machine Learning Society that is the institutional body responsible for running the conference ICML 2020, or to the Institutional Review Board of Carnegie Mellon University (contact via irb-review@andrew.cmu.edu).

**Funding:** This work was supported in part by National Science Foundation CAREER award 1942124 and in part by National Science Foundation Communication and Information Foundations 1763734. NSF CAREER award 1942124 was awarded to Nihar Shah (https://www.nsf.gov/awardsearch/showAward?AWD_ID=1942124&HistoricalAwards=false) NSF CIF 1763734 was awarded to Nihar Shah (https://www.nsf.gov/awardsearch/showAward?AWD_ID=1763734&HistoricalAwards=false) The funders had no role in study design, data collection and analysis, decision to publish, or preparation of the manuscript.

**Competing interests:** The authors have declared that no competing interests exist.

to ensure that the process is designed in a principled manner and does not result in unintended consequences such as bias or inefficiency [1]. However, a long line of work identifies various shortcomings of the review system, such as bias in review decisions based on authors' and reviewers' identities [2–6], commensuration biases [7, 8], miscalibration [9, 10], positive-outcome bias [11, 12], strategic behavior [13, 14], other unwanted social influences and human cognitive biases in the decision-making process [15–17], and ultimately the lack of consistency in review outcomes [18–20]. This motivates research effort and policy changes that aim at improving peer review. In this work, we continue the effort on scrutinizing various parts of the review system and report the results of a large-scale randomized experiment on the discussion stage of scientific peer review.

## 1.1 Discussion stage in scientific peer review

In many journals (e.g., Nature and PNAS) reviewers do not communicate with each other, and decisions are made by editors based on independent opinions of reviewers. In contrast, many conferences (which in computer science are considered to be a final destination for research and typically ranked higher than journals) and grant committees (e.g., NSF) implement an additional discussion stage that takes place after reviewers submit their initial independent reviews. The purpose of the discussion stage is to allow reviewers to exchange their opinions about the submission and correct each other's misconceptions. As a result of the discussion, reviewers are supposed to reach a consensus on the submission or boil their disagreement down to concrete arguments that can later be evaluated by chairs of the selection committee.

Given that independent opinions of reviewers often demonstrate a substantial amount of disagreement [21–24], the discussion stage may seem to be an appealing opportunity to reduce the load on editors and chairs by letting reviewers resolve their disagreements themselves. However, the aforementioned studies also demonstrate that while discussion increases the within-group agreement, the agreement between two groups of reviewers participating in parallel discussions of the same submission does *not* improve (and often the agreement across groups decreases after within-group discussions). This finding hints that reviewers within the group reach a consensus not because they identify the "correct" decision, but due to some other unknown artifacts of group discussion. The work [24] furthers this investigation, by qualitatively analysing the group discussions. They find that discourse around score calibration of reviewers plays a pivotal role in the variability in outcomes across different groups. These findings imply that group discussions may lead to a false sense of reliability on a groups' final decision. More generally, this observation agrees with a long line of research in psychology [25–29] which demonstrates that decisions stemming from a *group discussion* are susceptible to various biases related to social influence, such as confirmation bias, distortion in judgment or compliance due to social pressures based on personal characteristics.

Importantly, lack of reliability in peer-review discussions may have far-reaching consequences not just for a particular submission, but also for career trajectories of researchers due to the widespread prevalence of the Matthew effect ("rich get richer") in academia [30, 31]. Thus, in this work, we focus on investigating a cause of lack of reliability in peer review discussions and specifically focus on examining the presence of herding behaviour.

## 1.2 Herding behaviour

We consider a specific manifestation of social influence that results in *"herding behaviour"*— an effect when agents are doing what others are doing rather than choosing the course of actions based on the information available to them [32, 33]—in peer-review discussions. A long line of work on human decision-making establishes the presence of various biases related

to the first piece of information received by an individual. Seminal work by Tversky and Kahneman [34] introduced the anchoring-and-adjustment heuristic, which suggests that after being anchored to an initial hypothesis, humans tend to adjust insufficiently because adjustments are effortful and tend to stop once a plausible estimate is reached. This heuristic is also known as anchoring bias, studied in [35–39]. Another line of work studies the effect of the order of information provision on information retention [40], known as primacy and recency effects. Several human subject studies [41, 42] find that participants remember information better when they appear at the beginning or at the end of a learning episode.

In particular, these works show that an initial signal received by a decision-maker (even when being clearly unrelated to the underlying task) often has a disproportionately strong influence on the final decision. Projecting this observation on the group discussion setting, we study whether the first argument made in the discussion may exert an undue influence on the opinions of subsequent contributors, thereby leading to herding behaviour.

Past literature on group decision making [43–45] indeed suggests that the herding behaviour (titled "first advocacy" effect in these works) is present in group discussions. We discuss these past works in relation to our work in more detail in S4 Appendix. In [43] it is observed that the first solution proposed to a group predicts the group decision better than an aggregate of initial opinions independently expressed in a pre-discussion survey. The work [44] documents an impact of the interplay between the status of discussion participants and the opinion of the group member who proposed the first concrete solution on the final group decision. Closest to the present work, [45] further investigates the herding effect in a semi-randomized controlled trial and declares that the initiators of discussion manage to influence the group opinion when they step in after an initial general discussion of the problem, that is, when they have some understanding of the general opinions of other discussants, but no concrete decisions have been proposed.

*With this motivation, in this work we analyze the presence of the herding effect in the discussion stage of peer review*. To formalize the research question, we note that in the current review system it is often the job of the discussion chair (area chair in conferences, committee chair in grant proposal review) to maintain the order in which reviewers speak up in the discussion. In the absence of a standardized approach, some chairs may call upon the reviewer whose initial opinion is the most extreme to initiate the discussion, others may request the most positive or most negative reviewer to start. Another option for the discussion chair is to initiate the discussion themselves or to choose an initiator based on their seniority or expertise. In the presence of herding, the uncertainty in the choice of the strategy may impact the outcome of a paper or a grant proposal (which becomes dependent on the essentially arbitrary choice of the discussion initiator by the chair), thereby increasing the undesirable arbitrariness of the process. With the above motivation, in this work we aim at testing the presence of the herding effect in peer-review discussions, investigating the following research question:

*Research Question*: Given a set of reviewers who participate in the discussion of a submission, does the final decision for the submission causally depend on the choice of the discussion initiator made by the discussion chair?

## 2 Methods

In this section, we outline the design of the experiment we conducted to investigate the research question of this paper.

## 2.1 Setting of the experiment

The experiment was conducted in the peer-review process of ICML 2020—a flagship machine learning conference that receives thousands of paper submissions and manages a pool of thousands of reviewers. The ICML peer review is organized in a double-blind manner and, similar to most other top machine learning and artificial intelligence conferences, follows the timeline outlined in Fig 1.

Given our focus on the discussion dynamics, we describe the discussion process of ICML 2020 in more detail. During the discussion, reviewers (typically three or four per paper) and area chairs (one per paper; the role of area chairs is equivalent to that of associate editors in journal peer review) have access to the author feedback and are able to asynchronously communicate with each other (but not with authors) via a special online interface. The discussion between reviewers is anonymous (i.e., reviewers do not see each other's names), but area chairs know identities of reviewers and vice versa. For the papers assigned to them, each reviewer is expected to carefully read the author rebuttal as well as the reviews written by the other reviewers, and participate in the discussion.

## 2.2 Idea of the experiment

To investigate a *causal* relationship between the choice of the discussion initiator and the outcome of a paper, the experiment we design in this work follows an A/B testing pipeline. Specifically, the set of papers involved in the experiment is split into two groups that receive different treatments, where the treatments are designed such that the difference in some observable outcomes of papers across groups is indicative of the presence of herding. For the reasons that are clarified below, we refer to the two groups of papers as $\mathcal{P}_+$ and $\mathcal{P}_-$.

The key challenge of this work is to design appropriate treatments and we begin by specifying requirements that the treatment scheme must satisfy. First, we intuitively expect herding (if present) to move the outcome of a discussion towards the opinion of the discussion initiator. Thus, to achieve a high detection power, we induce the following requirement:

> **Requirement 1**: The treatment scheme should induce a difference across two groups of papers in terms of the initial opinion of reviewers who initiate discussions within these groups. That is, in $\mathcal{P}_+$, reviewers with a positive initial opinion should initiate discussion more often while in $\mathcal{P}_-$, reviewers with a negative initial opinion should be more active in initiating discussion.

Moving on to the second requirement, we note that not every treatment scheme that satisfies Requirement 1 is valid for testing our research question. Indeed, one idiosyncrasy of conference peer review (at least in the machine learning and artificial intelligence areas) is that some reviewers may choose to ignore the discussion. In fact, the analysis of the review process

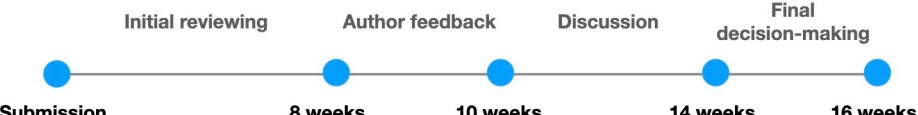

**Fig 1. Timeline of the peer-review process of typical machine learning and artificial intelligence conferences.** Upon the release of initial reviews, authors of papers have several days to write a response to reviewers, followed by the discussion stage. Finally, program chairs aggregate the results of the review process into final decisions. The duration of each stage varies across conferences, and this figure corresponds to the ICML 2020 review process with the duration of each stage rounded to weeks.

of another leading machine learning conference NeurIPS 2016 [46] revealed that only 30% of 13,674 paper-reviewer pairs had a message posted by the reviewer in the associated discussion, showing that the set of discussion participants is generally a strict (and somewhat small) subset of reviewers assigned to the paper. Thus, in the conference peer-review setting, any intervention that impacts *the order* in which reviewers join the discussion may also impact *the population* of reviewers who participate in the discussion, thereby introducing a confounding factor in our analysis. To mitigate this issue, we introduce another requirement that must be satisfied by the treatments:

> **Requirement 2**: The treatment scheme should not introduce any difference across two groups of papers other than in the opinion of the discussion initiator. That is, distributions of the participating reviewers and other characteristics of the discussion should be the same across $\mathcal{P}_+$ and $\mathcal{P}_-$.

We note that in some settings (e.g., panel discussion in grant proposal review where all reviewers are required to participate) Requirement 2 (at least its part about the distribution of participating reviewers) is naturally met. However, as our experiment is implemented in conference peer review, we impose this requirement to remove associated confounding factors.

## 2.3 Treatment scheme

Keeping Requirements 1 and 2 in mind, we now introduce the treatment scheme that we use in the experiment. The key component of our treatment scheme is a proxy towards the opinions of discussion participants. For this, recall that discussions begin *after* the initial reviews are submitted. Thus, we use overall scores given in the initial reviews to identify reviewers with the most positive and the most negative initial opinions about the paper. With these preliminaries, we execute the following treatments:

- Treatment for the positive group of papers $\mathcal{P}_+$

  - *Step 1* We ask a reviewer with the most <u>positive</u> initial opinion about the paper to *initiate* the discussion.

  - *Step 2* We then ask a reviewer with the most <u>negative</u> initial opinion to *contribute* to the discussion.

- Treatment for the negative group of papers $\mathcal{P}_-$

  - *Step 1* We ask a reviewer with the most <u>negative</u> initial opinion about the paper to *initiate* the discussion.

  - *Step 2* We then ask a reviewer with the most <u>positive</u> initial opinion to *contribute* to the discussion.

Both treatments consist of two steps: in the first step, a reviewer with an extreme opinion about the paper is asked to initiate the discussion. In the second step, the reviewer whose opinion is on the another extreme of the spectrum is asked to contribute to the discussion. Importantly, both treatments proceed to Step 2 (after some waiting time) even if the reviewer requested to initiate the discussion in the first step fails to fulfil the request.

Let us now discuss the design of the treatment scheme in light of the requirements we formulated earlier. First, observe that the order in which reviewers with the most positive and the most negative initial opinions about the paper are approached is different across treatments. Provided that a sufficiently large fraction of reviewers comply with our requests, we should

expect to see a difference in opinions of discussion initiators across $\mathcal{P}_+$ and $\mathcal{P}_-$ as stated in Requirement 1. Second, we note that Step 1 of the treatments alone may induce a difference in the population of reviewers who *participate* in the discussion across $\mathcal{P}_+$ and $\mathcal{P}_-$. To remove this confounding factor, both treatments implement Step 2 that is designed to "balance" the impact of Step 1, thereby aiming to satisfy Requirement 2.

Overall, we designed the intervention with a goal of satisfying the aforementioned requirements. However, a priori we cannot guarantee that these requirements are indeed satisfied in the real experiment. For example, Requirement 2 is not satisfied if the balancing part (Step 2) of the treatments fails to equalize the populations of participating reviewers across conditions. To further support our experimental methodology, in Section 3 we empirically check for satisfaction of the stated requirements.

As a final remark, we note that ideally, we would like to fully control the choice of the discussion initiator for each paper, thereby implementing the conventional randomized controlled experiment. However, in the conference peer-review setup, organizers have only limited ability to impact the behaviour of reviewers. Thus, our treatment scheme follows the concept of the randomized encouragement design [47], in which the treatments are not enforced, but encouraged.

## 2.4 Details of the experiment

Let us now clarify some important aspects of the experiment.

- **Sample size** In 2020, ICML received more than 5,000 paper submissions out of which 4,625 papers remained in the process (i.e., were not withdrawn) at the time when the discussion period began. While we would like to run the experiment using all these papers, we note that some reviewers may be the most positive or the most negative reviewers for multiple papers, being overburdened with requests to initiate (contribute to) the discussion of these papers. To limit the additional load on reviewers induced by our experiment, for each reviewer we limit the number of papers the reviewer is asked to initiate the discussion or contribute to the discussion to one each (in total, each reviewer may receive at most two requests). This condition puts the limit on the number of papers we can use in the experiment. Consequently, to compensate for the potential decrease of power, we focus the scope of the experiment on the *borderline* papers with some disagreement between reviewers as we expect the effect (if any) to be the most prominent in these papers. As a result, we end up with a pool $\mathcal{P}$ of 1,544 papers used in the experiment that we split into $\mathcal{P}_+$ and $\mathcal{P}_-$ uniformly at random subject to the aforementioned constraint on the additional reviewer load (see criteria for borderline papers and other details in S2 Appendix). The experiment also involved 2,797 reviewers who participated in the discussion of at least one paper from the experimental pool.

- **Implementation of the treatment scheme** To ensure that the behaviour of area chairs and reviewers is not altered by awareness of the experiment, we implement the treatment scheme at the level of program chairs and do not notify other committee members about the experiment. Specifically, the requests to initiate or contribute to the discussion were sent over email on behalf of the program chairs. To further maximize the power of our test, we first open the discussion interface without notifying the general pool of reviewers, and implement Step 1 of both treatments, giving reviewers more time to fulfil our request.

- **Statistical analysis** We employ two-sided permutation test [48] to test for difference across groups $\mathcal{P}_+$ and $\mathcal{P}_-$. Specifically, the analysis is conducted at the level of papers and we compute *p* values over 10,000 permutations of papers across conditions.

- **Data and code availability** We note that the release of experimental data would compromise the reviewers' confidentiality. Thus, following prior works that empirically analyze the conference peer-review process [3, 18, 46], and complying with the conference's policy, we are unable to release the data and code from the experiment.

- **Avoiding conflict of interests** Three members of the study team were involved in the ICML decision-making process. Nihar Shah served as an area chair and Aarti Singh and Hal Daumé III were program chairs. To avoid the conflict of interests, Nihar Shah, Aarti Singh and Hal Daumé III were not aware of what papers were used in the experiment. Moreover, we excluded papers chaired by Nihar Shah from the analysis.

- **Ethics statement** Finally, we note that if the herding effect is present in ICML discussions, our intervention may place some papers ($\mathcal{P}_-$) at a disadvantage, while giving an unfair advantage to other papers ($\mathcal{P}_+$). We carefully considered this risk when designing the experiment and concluded that it does not exceed the risk that is otherwise present in the review process. Indeed, currently, there is no standardized approach towards choosing discussion initiators and it is often the job of the discussion chair to maintain the order in which reviewers speak up in the discussion. Different discussion chairs implement different strategies and, under the presence of herding, this variability results in randomness in decisions. Hence, even if herding is present, the impact of our intervention would not go beyond the unfairness that is otherwise present in the review system. This study was analyzed by Carnegie Mellon University's Institutional Review Board that agreed with our reasoning and approved the study.
  Additionally, to avoid the Hawthorne effect, we employ deception and do not notify reviewers about the experiment or collect consent. The deception was approved by Carnegie Mellon University's Institutional Review Board who issued a waiver of informed consent from the participants. We debriefed all participants after the end of the review process.

  More details on the experiment design and exact schedule of our intervention are given in S2 Appendix.

## 3 Results of the experiment

In this section, we present the results of the experiment. First, we empirically check whether our treatment scheme satisfies requirements formulated in Section 2. In that, we begin with some general statistics on the discussion process to check satisfaction of Requirement 2 (Section 3.1). We then discuss the efficacy of the intervention we employed (Section 3.2) and confirm that Requirement 1 is well-satisfied. Finally, we conclude with the analysis of the research question we study in this work (Section 3.3). For brevity, for any paper, we use $R_+$ (respectively, $R_-$) to refer to the reviewer with the most positive (respectively, negative) initial opinion about the paper as determined by the overall scores given in the initial reviews.

### 3.1 Preliminary analysis (data to check for satisfaction of Requirement 2)

We begin with providing data-based evidence which lets us verify whether our treatment scheme satisfies Requirement 2, that is, does not introduce differences across $\mathcal{P}_+$ and $\mathcal{P}_-$ in characteristics other than the opinion of the discussion initiator. For this, Table 1 provides some comparative statistics on the discussion process for the papers involved in the experiment. Overall, we note that many important parameters of the discussion are similar across the two conditions. This observation provides quantitative evidence that the randomization of papers to conditions occurred successfully and Requirement 2 is satisfied.

**Table 1. Comparative statistics on the discussion process.**

| | $\mathcal{P}_+$ | $\mathcal{P}_-$ |
|---|---|---|
| 1. Number of papers | 755 | 789 |
| 2. Mean initial score (all revs) | 3.52 | 3.52 |
| 3. Standard deviation of initial scores (all revs) | 1.12 | 1.11 |
| 4. Mean initial score (revs in discussion) | 3.44 | 3.46 |
| 5. Percentage of papers with active discussion | 97% | 97% |
| 6. Mean number of discussion participants (revs + area chairs) | 3.14 | 3.06 |
| 7. Mean discussion length (# messages) | 4.41 | 4.24 |
| 8. Percentage of papers with $R_+$ active in discussion | 79% | 79% |
| 9. Percentage of papers with $R_-$ active in discussion | 87% | 84% |

Table notes: Comparison of some discussion statistics between two groups of papers ($\mathcal{P}_+$ and $\mathcal{P}_-$) receiving different treatments. Except Row 4, all values are computed using all papers including those with no discussion. Permutation test at the level 0.05 (two-sided; before multiple-testing adjustment) with 10,000 iterations does not reveal significant differences between conditions in any of the criteria.

Looking closer at the most relevant parameters, we first focus on Rows 2 and 4 of that compare mean overall scores (the overall score takes integer values from 1 to 6 where larger values indicate higher quality) given by reviewers in the initial reviews, that is, before reviewers got to see the other reviews and the author feedback. Interestingly, mean initial scores given by reviewers who participate in the discussion (Row 4) appear to be lower than mean scores computed over all reviewers (Row 2), suggesting that those who give lower scores are more active in discussing papers (see also Rows 8 and 9). However, there is no significant difference between two groups of papers ($\mathcal{P}_+$ and $\mathcal{P}_-$) in these values. Hence, the data indicates that this trend is independent of the choice of the treatment as requested by Requirement 2.

Next, the activity of reviewers with the most positive (respectively, negative) initial opinion in the discussion (Rows 8 and 9) is similar across the two groups of papers. This observation supports the intuition that our treatment scheme does not introduce a difference across conditions in the distributions of reviewers who participate in the discussion. Finally, we note that most of the papers used in the experiment had some discussion (Row 5). Specifically, we see that the mean number of participants in a paper's discussion and the length of a paper's discussion is similar across the two conditions, where we measure the length as the number of messages in a paper's discussion thread (Rows 6 and 7). With this observation, we conclude this section and note that data reported in Table 1 supports our treatment scheme in light of Requirement 2.

## 3.2 Efficacy of the intervention (data to check for satisfaction of Requirement 1)

In the previous section we confirmed that our intervention did not introduce a difference across conditions in metrics such as intensity of discussions and the population of participating reviewers. This observation suggests that our treatment scheme satisfied Requirement 2 and indicates the appropriateness of our intervention. However, in order for the experiment to have sufficient power to detect the effect, the intervention needs to satisfy Requirement 1 and introduce a difference across conditions in the order in which reviewers join the discussion of the papers. Indeed, if all the emails we sent to reviewers were ignored (i.e., our attempt to impact the order failed), the subsequent analysis will not detect the phenomena even when the phenomena is present.

**Table 2. Does the intervention affect who initiates the discussion?.**

| | $\mathcal{P}_+$ | $\mathcal{P}_-$ | Δ | Δ 95% CI | p value |
|---|---|---|---|---|---|
| 1. Mean initial score (initiator) | 4.03 | 2.76 | 1.27 | [1.15, 1.39] | < .001 |
| 2. Percentage of discussions initiated by $R_+$ | 53% | 9% | 0.44 | [0.39, 0.48] | < .001 |
| 3. Percentage of discussions initiated by $R_-$ | 15% | 59% | −0.44 | [−0.48, −0.39] | < .001 |

Table notes: The impact of the intervention on who initiates the discussion. To compute values for Row 1, we use 1,140 papers for which (i) the discussion was initiated, and (ii) the discussion initiator was a reviewer (and not the area chair). For the last two rows, we use all papers including those with no discussion. Bootstrapped confidence intervals are constructed for the difference of the relevant quantities between conditions. All p values for the difference between $\mathcal{P}_+$ and $\mathcal{P}_-$ are two-sided and computed using the permutation test with 10,000 iterations.

Table 2 reports relevant statistics and indicates a large difference between positive and negative groups of papers, suggesting that our intervention did impact the order in which reviewers joined the discussion. Indeed, Rows 2 and 3 show that more than half of discussions in the positive group $\mathcal{P}_+$ were initiated by reviewers with the most positive initial opinion and only 15% were initiated by reviewers with the most negative initial opinion. Conversely, in the negative group $\mathcal{P}_-$, reviewers with the most negative initial opinion initiated the discussion a lot more often than reviewers with the positive initial opinion. This asymmetry results in a considerable difference of scores given by discussion initiators in their initial reviews (Row 1). Overall, Table 2 suggests that our treatment scheme satisfied Requirement 1. Coupled with a large sample size, this observation ensures that our experiment has a strong detection power.

### 3.3 Main causal analysis

Having confirmed that the intervention we implemented in the experiment reasonably satisfies the necessary requirements, we now continue to the analysis directly related to the research question we study in this work. Specifically, as we explained in the introduction and in Section 2, if herding behaviour exists, we expect it to manifest in the final decisions being disproportionately influenced by the opinion of the discussion initiator. Hence, given that for the positive group of papers $\mathcal{P}_+$ the initial opinion of the discussion initiator was on average significantly more positive than that of initiators of discussions for the negative group of papers $\mathcal{P}_-$, we expect (if herding exists) to observe a disparity in the eventual acceptance rates between conditions.

To understand the effect of the experiment, we first note the observed mean final scores in different sets of reviewers corresponding to groups $\mathcal{P}_+$ and $\mathcal{P}_-$. Among the reviewers who initiated the discussion, the mean final scores were 4.13 and 2.96 respectively. Among the reviewers who participated in the discussion, the mean final scores were 3.53 and 3.53, and among all reviewers, the mean final scores were 3.47 and 3.48 in $\mathcal{P}_+$ and $\mathcal{P}_-$ respectively.

Table 3 formalizes the intuition and performs the comparison of acceptance rates across papers that received different treatments (Row 1). Additionally, Table 3 displays the updates of the scores made by reviewers (Rows 2–5). Based on the presented data, we make two observations:

- First, the data does not indicate a statistically significant difference between acceptance rates in the two groups of papers ($\mathcal{P}_+$ versus $\mathcal{P}_-$). Thus, despite discussion initiators had considerably different initial opinions about papers from $\mathcal{P}_+$ and $\mathcal{P}_-$, the outcomes of the discussion were distributed similarly across conditions.

**Table 3. Does the intervention affect the outcome of the papers?.**

| | $\mathcal{P}_+$ | $\mathcal{P}_-$ | Δ | Δ 95% CI | *p* value | Cohen's d |
|---|---|---|---|---|---|---|
| 1. Acceptance rate | 0.21 | 0.25 | −0.04 | [−0.08, 0.01] | 0.104 | 0.08 |
| 2. Change in mean score (initiator) | −0.10 | 0.20 | −0.30 | [−0.37, −0.23] | < .001 | 0.47 |
| 3. Change in mean score (all revs) | 0.01 | 0.01 | 0.00 | [−0.03, 0.04] | .949 | 0.00 |
| 4. Change in mean score (revs in discussion) | 0.03 | 0.02 | 0.01 | [−0.04, 0.06] | .697 | 0.02 |
| 5. Change in standard deviation of scores (all revs) | −0.23 | −0.21 | −0.02 | [−0.05, 0.02] | .296 | 0.05 |

Table notes: The impact of the intervention on the final outcome of papers. For Row 2, we use 1,140 papers for which (i) the discussion was initiated, and (ii) the discussion initiator was a reviewer (and not the area chair). For Row 4, we use papers with discussion. For all other rows, we use all papers including those with no discussion. Bootstrapped confidence intervals are constructed for the difference of the relevant quantities between conditions. The *p* value for the significance in difference between $\mathcal{P}_+$ and $\mathcal{P}_-$ in Row 1 is obtained using Fisher's exact test [48] (applicable for binary outcomes). The remaining *p* values for the difference between $\mathcal{P}_+$ and $\mathcal{P}_-$ are two-sided and computed using the permutation test with 10,000 iterations. In the last column, Cohen's d value indicates the effect size of the difference between the two groups, where d value of 0.2 and 0.5 is interpreted as small and medium effect size respectively.

- Second, the data on the score updates suggests that in their final evaluations, reviewers tend to converge to the mean of initial independent opinions irrespective of the treatment we applied to a paper. Indeed, Row 2 demonstrates that the initiators of discussions update the scores towards the mean of all initial scores. Next, Rows 3 and 4 show that a significant update made by the discussion initiators is compensated by the update made by other reviewers, such that the overall amount of change in the mean scores is negligible. As expected, the outlined dynamics result in a significant decrease in the variance of scores per paper, but the effect is the same for both groups of papers (Row 5).

Overall, we find no evidence of herding in the discussion phase of ICML 2020 peer review. We provide supporting information about the analysis in S3 Appendix.

## 4 Discussion

Past work has documented an undue influence of the first piece of information on the final decision [32, 34, 38–40] in various other settings and applications. Our experiment aimed at identifying the presence of herding behaviour in the peer-review discussions of the ICML 2020 conference. Our intervention successfully managed to achieve an imbalance in the opinion of the discussion initiators across conditions. However, the difference in the acceptance rates across conditions is not statistically significant, and the change in the mean scores of all the reviewers and the set of reviewers that participated in the discussion, before and after the discussion, is small. We thus find no evidence of herding in peer review.

In this work, we focused on the manifestation of herding effect in peer-review discussions. In future work, it is of interest to understand whether the separate cognitive biases attributed to herding behavior such as anchoring bias [35–39] and, primacy and recency effect [40] have any individual role to play in peer-review discussions.

Regarding policy implications, the absence of the effect suggests that the absence of a unified approach towards discussion management does not result in an increased arbitrariness of the resulting decisions. Thus, the question of identifying the source of the spurious agreement between peer reviewers observed in past works [21–24] remains open.

Finally, we urge the reader to be aware of the caveats that we listed in S1 Appendix. These caveats present limitations of the study, and are found in the design of the intervention, the choice of papers for the experiment, the satisfaction of Requirement 2, the spurious

correlations induced by reviewer identity, opinion of the discussion initiator and the possibility of alternative models for herding. While we do not believe that any of these caveats affected the outcome of this experiment in a significant way, they may be important for the design of follow-up studies.

## Supporting information

**S1 Appendix. Caveats regarding the design and analysis of the experiment** [3, 17, 18, 46, 49].
(PDF)

**S2 Appendix. Additional details on the experiment.** Fig 2. Timeline of the experiment. Day X is the day of the official discussion opening.
(PDF)

**S3 Appendix. Additional analysis.**
(PDF)

**S4 Appendix. Relation to past work on group discussion** [17, 21–24, 43–45, 50–52].
(PDF)

## Acknowledgments

We thank Edward Kennedy for useful comments on the initial design of this study. We are also grateful to the support team of the Microsoft Conference Management Toolkit (CMT) for their continuous support and help with multiple customization requests. We thank Christopher Baethge for pointing out that the earlier version of this work had a slightly larger value in Row 1 of Table 3 due to our use of the permutation test in the earlier version. Finally, we appreciate the efforts of all reviewers and area chairs involved in the ICML 2020 review process.

## Author Contributions

**Conceptualization:** Ivan Stelmakh, Charvi Rastogi, Nihar B. Shah, Aarti Singh, Hal Daumé, III.

**Data curation:** Ivan Stelmakh, Aarti Singh, Hal Daumé, III.

**Formal analysis:** Ivan Stelmakh.

**Funding acquisition:** Nihar B. Shah, Aarti Singh.

**Investigation:** Ivan Stelmakh, Nihar B. Shah.

**Methodology:** Ivan Stelmakh, Charvi Rastogi, Nihar B. Shah, Aarti Singh, Hal Daumé, III.

**Project administration:** Nihar B. Shah.

**Supervision:** Nihar B. Shah, Hal Daumé, III.

**Visualization:** Ivan Stelmakh.

**Writing – original draft:** Ivan Stelmakh, Nihar B. Shah, Hal Daumé, III.

**Writing – review & editing:** Ivan Stelmakh, Charvi Rastogi, Nihar B. Shah, Aarti Singh, Hal Daumé, III.

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
