## [Decision Letter · Decision Letter 0]

29 Nov 2022

PONE-D-22-17535

A Large Scale Randomized Controlled Trial on Herding in Peer-Review Discussions

PLOS ONE

Dear Dr. Ivan Stelmakh,

Thank you for submitting your manuscript to PLOS ONE. After careful consideration, we feel that it has merit but does not fully meet PLOS ONE’s publication criteria as it currently stands. Therefore, we invite you to submit a revised version of the manuscript that addresses the points raised during the review process.

We have received two reports. One reviewer recommends a minor, while the other one recommends a major revision. Please, pay attention to every point raised by the referees.

A rebuttal letter that responds to each point raised by the academic editor and reviewer(s). You should upload this letter as a separate file labeled 'Response to Reviewers'.A marked-up copy of your manuscript that highlights changes made to the original version. You should upload this as a separate file labeled 'Revised Manuscript with Track Changes'.An unmarked version of your revised paper without tracked changes. You should upload this as a separate file labeled 'Manuscript'

We look forward to receiving your revised manuscript.

Kind regards,

Jaume Garcia-Segarra

Academic Editor

PLOS ONE

“This work was supported in part by National Science Foundation CAREER award 1942124 and in part by

National Science Foundation Communication and Information Foundations 1763734”

“This work was supported in part by National Science Foundation CAREER award 1942124 and in part by National Science Foundation Communication and Information Foundations 1763734.

NSF CAREER award 1942124 was awarded to Nihar Shah (https://www.nsf.gov/awardsearch/showAward?AWD_ID=1942124&HistoricalAwards=false)

NSF CIF 1763734 was awarded to Nihar Shah (https://www.nsf.gov/awardsearch/showAward?AWD_ID=1763734&HistoricalAwards=false)

4. We noted in your submission details that a portion of your manuscript may have been presented or published elsewhere. [DETAILS AS NEEDED] Please clarify whether this [conference proceeding or publication] was peer-reviewed and formally published. If this work was previously peer-reviewed and published, in the cover letter please provide the reason that this work does not constitute dual publication and should be included in the current manuscript.

Reviewers' comments:

Reviewer's Responses to Questions

**Comments to the Author**

1. Is the manuscript technically sound, and do the data support the conclusions?

Reviewer #1: Yes

Reviewer #2: Partly

2. Has the statistical analysis been performed appropriately and rigorously? 

Reviewer #1: Yes

Reviewer #2: Yes

3. Have the authors made all data underlying the findings in their manuscript fully available?

Reviewer #1: No

Reviewer #2: No

4. Is the manuscript presented in an intelligible fashion and written in standard English?

Reviewer #1: Yes

Reviewer #2: Yes

5. Review Comments to the Author

Reviewer #1: The paper addresses the topic of the peer review process and how discussions among reviewers might lead to cognitive biases. The authors focus on the peer review process of conferences and grant committees because in these cases a discussion between reviewers is part of the process compared to journals.

Overall, I found the article well written and interesting. I would suggest a few changes to strengthen the manuscript.

My biggest concern with the manuscript is the use of psychological terminologies. In my opinion, the experiment described in the manuscript does not investigate herding behavior but more specifically the possibility of an anchoring effect in the peer review process of conferences. This becomes also clear by the literature that is cited. Tversky and Kahneman, 1974; Strack and Mussweiler, 1997; Mussweiler and Strack, 2001 are all articles investigating the anchoring effect and not herding behavior. I would strongly advice the authors to revise the manuscript to be more accurate with the terminology.

I would also recommend looking further into literature on the primacy and recency effect. Especially with the design of the experiment, the authors should consider that in addition to an anchoring effect there might possibly be a primacy or recency effect. It would also be interesting to examine the length of the discussion although I am not sure if the authors have access to this information.

The recency effect might be a possible explanation for the update towards the mean of the initiators (table 3 row 2).

Although I understand the need for requirement 2 from an experimental perspective, I am not sure if requirement 2 might be the reason why the results do not show evidence for an anchoring effect. Requirement 2 ensured that the discussion is balanced and that the only difference between treatments is the initiators opinion. This might not necessarily reflect the reality of the discussion process. In a disussion where the initiators opinion is dominating, an anchoring effect might occur. The authors should include this fact in the discussion.

A minor comment concerns the explanation of shortcomings of the review system in the introduction. In my opinion the authors should elaborate a bit more to better introduce the aim of their research.

Reviewer #2: The authors carried out a singular study to examine the presence of herding behavior in discussion stage in scientific peer-review. The relevance of this study is justified because of the peer review is a crucial step for publishing research works in the most important scientific journals. To guarantee that the process is rigorous and is supported in a principled manner, it is necessary to show that the final decision of the peer review is not biased. That is the main goal of this research. For this purpose, the authors carried out an experimental study. The set of papers involved in the experiment was split into 2 groups, balancing the order of positive and negative opinion that initiate the discussion of reviewers. The results showed no evidence of herding behavior in the discussion phase of peer review, but results showed a significant decrease in the variance of final scores due to the agreement among reviewers.

I appreciated having the opportunity to review this manuscript. I think the topic is interesting and relevant in matter of making decisions. However, there are some critical questions and comments that I would like to point out the authors to address, and the final decision should be pending of how the authors are handling the next changes.

Major Comments:

1. The introduction section is a little brief. In this section, I miss that the authors summarize deeper the literature of biases making decisions. For example:

a) On paragraph 3 of page 2, the authors wrote: “This finding hints that reviewers within the group reach a consensus not because they identify the “correct” decision, but due to some other artifacts of group discussion” but the authors did not indicate which are these other artifacts of groups discussion, and their implications.

b) On paragraph 3 of page 2, the authors wrote: “More generally, this observation agrees with a long line of research in psychology (Asch, 1951; Baron et al., 1996; Lorenz et al., 2011; Janis, 1982; Cialdini and Goldstein, 2004) which demonstrates that decisions stemming from a group discussion are susceptible to various biases related to social influence”, but the authors did not indicate which biases related to social influence are talking about and their effects.

c) Also, the authors talk about herding behavior without going into in depth, despite that the research is focus on this bias (see paragraph 5 of page 2).

Hence, I think that the authors should synthesize deeper the literature of biases making decisions (its results and conclusions), especially the herding behavior which is the target of this research.

2. A second critical comment that I would like to point out the authors is regarding to the method section. I think that the explanation of the experimental development is a little confuse and it should be reformulated, being more precise and shorter. It is important that be clear which are the experimental conditions and what each of them implies.

3. On the last paragraph of page 4, the authors wrote: “Overall, we designed the intervention with a goal of satisfying the aforementioned requirements. However, a priori we cannot guarantee that these requirements are indeed satisfied in the real experiment”. I think that this paragraph is questioning the reliability and the appropriateness of the experimental design if the authors are not able to guarantee the control of main aspects on their experiment is basing on. Hence, I suggest to the authors remove to the research data those cases that authors cannot guarantee that the experimental requirements are satisfied in the real experiment and repeat the analyses without those data. The authors can report the results with and without these data in order to check their effects.

4. How did the authors handle those cases where the reviewers decided to ignore the discussion phase of peer review, or in those cases where the reviewers that had to initiate the discussion decide no initiate the discussion? Did those cases included in the study data? because I think that those cases no satisfy the requirements...

5. In the section of results, the authors checked firstly whether the requirement 2 was satisfied and then checked whether the requirement 1 was satisfied. As usually researchers answer their research questions in the order in which they are formulated, I think that the authors should check if the requirements are satisfied in the order in which they are proposed in the method section. This is just for a reason of thoroughness and order.

6. I think that the initial and final mean score of initiator, all reviewers, and reviewers in discussion should be reported on Table 3 in addition to the change in those punctuations. That information allows to the readers calculate the change in the punctuations, in order to be more transparent with the data, and to understand better the results. Also, I think that the p value of the changes in those punctuations per every experimental group (P+ and P-) should be reported on Table 3 to check if there were significant differences and, consequently, a herding effect within each experimental group, and not only between experimental groups.

7. Regarding to the two observations that the authors indicated on results section (see paragraphs 3 and 4 of page 8), any of them actually answer the research question of the paper. In my own opinion, to answer that question, authors should compare the initial and the final scores of reviewers within each experimental group not between groups. The conclusions that the authors point out are additional conclusions from the results.

8. Finally, the discussion section should be extended. I think this section should indicate the main goal of the study, include a brief summary of the main results and conclusions, the theoretical and practical implications of the findings, the main limitations of the study, and some suggestions for future research.

Minor Comments:

9. I suggest clarifying what mean NS, AS and HD (see paragraph 9 of page 5), because it is not reported in the paper.

10. In the section 3.1, the authors wrote: “Finally, we note that most of the papers used in the experiment had some discussion and the length of the discussion is similar across conditions (Rows 5 and 7)”, but the authors did not explain how the length of the discussion was measure. This is important to understand what meaning the length of the discussion (time spent on the discussion, paper extension of the discussion, …).

11. I suggest explaining the row 6 of table 1 in the results section.

6. PLOS authors have the option to publish the peer review history of their article (what does this mean?). If published, this will include your full peer review and any attached files.

Reviewer #1: No

Reviewer #2: No

---

## [Author Response · Author response to Decision Letter 0]

11 Jan 2023

We have provided our response to reviewers and editor requirements in an attached file named "Response to reviewers". We copy-paste the text from that file in this field, however, for readability we suggest the reviewers use the attached file. 

REVIEWER COMMENTS

Reviewer #1

The paper addresses the topic of the peer review process and how discussions among reviewers might lead to cognitive biases. The authors focus on the peer review process of conferences and grant committees because in these cases a discussion between reviewers is part of the process compared to journals.

Overall, I found the article well written and interesting. I would suggest a few changes to strengthen the manuscript.

My biggest concern with the manuscript is the use of psychological terminologies. In my opinion, the experiment described in the manuscript does not investigate herding behavior but more specifically the possibility of an anchoring effect in the peer review process of conferences. This becomes also clear by the literature that is cited. Tversky and Kahneman, 1974; Strack and Mussweiler, 1997; Mussweiler and Strack, 2001 are all articles investigating the anchoring effect and not herding behavior. I would strongly advice the authors to revise the manuscript to be more accurate with the terminology.

I would also recommend looking further into literature on the primacy and recency effect. Especially with the design of the experiment, the authors should consider that in addition to an anchoring effect there might possibly be a primacy or recency effect. It would also be interesting to examine the length of the discussion although I am not sure if the authors have access to this information.

The recency effect might be a possible explanation for the update towards the mean of the initiators (table 3 row 2).

Although I understand the need for requirement 2 from an experimental perspective, I am not sure if requirement 2 might be the reason why the results do not show evidence for an anchoring effect. Requirement 2 ensured that the discussion is balanced and that the only difference between treatments is the initiators opinion. This might not necessarily reflect the reality of the discussion process. In a discussion where the initiators opinion is dominating, an anchoring effect might occur. The authors should include this fact in the discussion.

A minor comment concerns the explanation of shortcomings of the review system in the introduction. In my opinion the authors should elaborate a bit more to better introduce the aim of their research.

Response: Thank you very much for your suggestions and feedback. We incorporated the suggestions in our manuscript, and the edits are highlighted in blue in the revised manuscript. We further provide our detailed response to the feedback here: 

In our work we consider the behavior of herding as defined in the seminal paper by Banerjee et. al. [1]. As mentioned in [1], herding behavior may be explained by many psychological internal mechanisms such as anchoring effect, primacy effect, recency effect. We discuss these possible internal mechanisms in more detail in our revised manuscript. However, it is important to note that our work focuses on observing the outcomes of herding behavior, and not on the internal mechanisms causing the herding behavior. Given the data available, we cannot determine the source of herding behavior, which could be any combination of anchoring, primacy and recency effects, and hence we only focus on the overall outcome of possible herding. 

Following your suggestion, we provide more discussion about the psychological internal mechanisms that can cause herding behavior in our revised manuscript. We reproduce the text here: 

“Seminal work by Tversky and Kahneman [30] introduced the anchoring-and-adjustment heuristic, which suggests that after being anchored to an initial hypothesis, humans tend to adjust insufficiently because adjustments are effortful and tend to stop once a plausible estimate is reached. This heuristic is also known as anchoring bias, studied in [31–35]. Another line of work studies the effect of the order of information provision on information retention [36], known as primacy and recency effects. Several human subject studies [37, 38] find that participants remember information better when they appear at the beginning or at the end of a learning episode.”

Thank you very much for the suggestion.

The review mentions that “it would be interesting to examine the length of the discussion”. In our analysis, we examine the length of discussion by measuring the number of separate messages in a paper’s discussion thread. This data is provided in Row 7 in Table 1. The text of the individual posts is not available, so we do not measure the length of each individual post in a discussion thread. 

Following your suggestion, we provide more information about the shortcomings of the review system in our introduction in the revised manuscript. We reproduce the text here for your reference: “However, a long line of work identifies various shortcomings of the review system [citations] such as bias in review decisions based on authors' and reviewers' identities, presence of unwanted social influences and human cognitive biases in the decision-making process, and ultimately the lack of consistency in review outcomes.”

Reviewer #2: 

The authors carried out a singular study to examine the presence of herding behavior in discussion stage in scientific peer-review. The relevance of this study is justified because of the peer review is a crucial step for publishing research works in the most important scientific journals. To guarantee that the process is rigorous and is supported in a principled manner, it is necessary to show that the final decision of the peer review is not biased. That is the main goal of this research. For this purpose, the authors carried out an experimental study. The set of papers involved in the experiment was split into 2 groups, balancing the order of positive and negative opinion that initiate the discussion of reviewers. The results showed no evidence of herding behavior in the discussion phase of peer review, but results showed a significant decrease in the variance of final scores due to the agreement among reviewers.

I appreciated having the opportunity to review this manuscript. I think the topic is interesting and relevant in matter of making decisions. However, there are some critical questions and comments that I would like to point out the authors to address, and the final decision should be pending of how the authors are handling the next changes.

Response: Thank you very much for your suggestions and feedback. We incorporated the suggestions in our manuscript, and the edits are highlighted in blue in the revised manuscript. We further provide our detailed response to each comment below.

Major Comments:

1. The introduction section is a little brief. In this section, I miss that the authors summarize deeper the literature of biases making decisions. For example:

a) On paragraph 3 of page 2, the authors wrote: “This finding hints that reviewers within the group reach a consensus not because they identify the “correct” decision, but due to some other artifacts of group discussion” but the authors did not indicate which are these other artifacts of groups discussion, and their implications.

Thank you for pointing that out. We add more discussion about the different artifacts of group discussion in our manuscript. We reproduce the text here: 

“This finding hints that reviewers within the group reach a consensus not because they identify the ``correct'' decision, but due to some other unknown artifacts of group discussion. The work [citation] furthers this investigation, by qualitatively analyzing the group discussions. They find that discourse around score calibration of reviewers plays a pivotal role in the variability in outcomes across different groups. These findings imply that group discussions may lead to a false sense of reliability on a groups' final decision.”

b) On paragraph 3 of page 2, the authors wrote: “More generally, this observation agrees with a long line of research in psychology (Asch, 1951; Baron et al., 1996; Lorenz et al., 2011; Janis, 1982; Cialdini and Goldstein, 2004) which demonstrates that decisions stemming from a group discussion are susceptible to various biases related to social influence”, but the authors did not indicate which biases related to social influence are talking about and their effects.

Thank you for pointing that out. We add more discussion about the different social influences in group discussions and their effects. We reproduce the text here: 

“More generally, this observation agrees with a long line of research in psychology [citations] which demonstrates that decisions stemming from a group discussion are susceptible to various biases related to social influence, such as confirmation bias, distortion in judgment or compliance due to social pressures based on personal characteristics.”

c) Also, the authors talk about herding behavior without going into in depth, despite that the research is focus on this bias (see paragraph 5 of page 2).

We add more discussion about the different psychological internal mechanisms behind herding bias in the manuscript. We reproduce the text here: 

“Seminal work by Tversky and Kahneman [citation] introduced the anchoring-and-adjustment heuristic, which suggests that after being anchored to an initial hypothesis, humans tend to adjust insufficiently because adjustments are effortful and tend to stop once a plausible estimate is reached. This heuristic is also known as anchoring bias, studied in [citations]. Another line of work studies the effect of the order of information provision on information retention [citation], known as primacy and recency effects. Several human subject studies [citation] find that participants remember information better when they appear at the beginning or at the end of a learning episode.”

 Further, we discuss the findings from other research on herding behavior in group discussions in more detail. We reproduce the text here: 

“In [citation] it is observed that the first solution proposed to a group predicts the group decision better than an aggregate of initial opinions independently expressed in a pre-discussion survey. The work [citation] documents an impact of the interplay between the status of discussion participants and the opinion of the group member who proposed the first concrete solution on the final group decision. Closest to the present work, [citation] further investigates the herding effect in a semi-randomized controlled trial and declares that the initiators of discussion manage to influence the group opinion when they step in after an initial general discussion of the problem, that is, when they have some understanding of the general opinions of other discussants, but no concrete decisions have been proposed.”

Hence, I think that the authors should synthesize deeper the literature of biases making decisions (its results and conclusions), especially the herding behavior which is the target of this research.

2. A second critical comment that I would like to point out the authors is regarding to the method section. I think that the explanation of the experimental development is a little confuse and it should be reformulated, being more precise and shorter. It is important that be clear which are the experimental conditions and what each of them implies.

We did not understand this comment by the reviewer. Could the reviewer please elaborate what was imprecise and confusing? Regarding the reviewer’s suggestion of being “shorter”: we tried to provide the details to be fully rigorous, so we do not understand what the reviewer wishes to condense or eliminate. The experimental conditions and implications are described under the heading “Treatment scheme” and further discussed under the “Details of the experiment” heading right after that. 

3. On the last paragraph of page 4, the authors wrote: “Overall, we designed the intervention with a goal of satisfying the aforementioned requirements. However, a priori we cannot guarantee that these requirements are indeed satisfied in the real experiment”. I think that this paragraph is questioning the reliability and the appropriateness of the experimental design if the authors are not able to guarantee the control of main aspects on their experiment is basing on. Hence, I suggest to the authors remove to the research data those cases that authors cannot guarantee that the experimental requirements are satisfied in the real experiment and repeat the analyses without those data. The authors can report the results with and without these data in order to check their effects.

4. How did the authors handle those cases where the reviewers decided to ignore the discussion phase of peer review, or in those cases where the reviewers that had to initiate the discussion decide no initiate the discussion? Did those cases included in the study data? because I think that those cases no satisfy the requirements…

We combine the responses to comments 3 and 4 since they are aimed at the same issue. 

In our analysis, we include the cases where the reviewers decided to ignore the discussion phase or where the reviewer asked to initiate the discussion did not initiate the discussion. The reason for our approach is avoiding selection bias, we explain this in more detail in the next paragraph. Literature in the field of causal inference has a lot of discussion on non-compliance and how ignoring non-complying data points (selecting only complying data points) can lead to biased estimates [1,2]. Our analysis approach is the same as the standard “Intention to treat” analysis commonly applied in causal inference literature [3].

While an analysis on data where the reviewers complied with the experiment design may increase the power of the test, it is very important to note that such an analysis may lead to selection bias and hence violate the false alarm guarantees of the statistical analysis. Specifically, it is possible that the reviewers who comply with the experiment are statistically significantly different in their behavior compared to the reviewers who do not comply with the experiment. If this is the case, then compliance will cause confounding in the analysis that you are suggesting. For instance, consider an example where senior reviewers are more likely to comply with our intervention by initiating the discussion. This could be because senior reviewers are more likely to attend to their reviewing responsibilities. In this example, if we observe herding behavior, it may be explained by the seniority of the discussion initiator if senior reviewers are more likely to be well-calibrated in their reviewing decisions. Hence, the analysis would show herding behavior even though the reason for the effect is not herding. 

Finally, the review mentions that these cases (where the reviewer did not comply with the experiment) do not satisfy the requirements of our experiment, however, that is not the correct interpretation of the requirements posed in the manuscript. The requirements as stated in our work are described for the two groups of papers created in our randomized controlled trial, and not for individual papers. Further, we show that our group-level requirements are satisfied in Table 2.

5. In the section of results, the authors checked firstly whether the requirement 2 was satisfied and then checked whether the requirement 1 was satisfied. As usually researchers answer their research questions in the order in which they are formulated, I think that the authors should check if the requirements are satisfied in the order in which they are proposed in the method section. This is just for a reason of thoroughness and order.

We followed the ordering based on the structure of the narrative of the results section. We consider “requirement 2” first as it is a preliminary requirement and doesn’t indicate the success of our main intervention, whereas “requirement 1” is concerned with the success of our main intervention. 

6. I think that the initial and final mean score of initiator, all reviewers, and reviewers in discussion should be reported on Table 3 in addition to the change in those punctuations. That information allows to the readers calculate the change in the punctuations, in order to be more transparent with the data, and to understand better the results. Also, I think that the p value of the changes in those punctuations per every experimental group (P+ and P-) should be reported on Table 3 to check if there were significant differences and, consequently, a herding effect within each experimental group, and not only between experimental groups.

We provide the difference between the initial and final mean scores of (i) initiators, (ii) reviewers and (iii) reviewers that participated in the discussion in Table 3. Further, we provide the initial mean scores of (i) initiators in Row 1 of Table 2, (ii) all reviewers in Row 2 of Table 1 and (iii) reviewers that participated in the discussion in Row 4 of Table 1. This information is provided in this order based on the narrative of the discussion of results. The final mean score for the three groups of reviewers can be straightforwardly deduced from this information.

Further, we provide the p value of the difference across groups in these changes (punctuations) in Rows 2,3 and 4 in Table 3. Our results show that while the difference is significant among the initiators (p value < 0.001), it is not significant for all reviewers or for reviewers that participated in the discussion. 

Finally, following your suggestion, we also highlight this in the concluding section. We reproduce the text here: 

“Further, we observe that within each condition, the change in the mean scores of all the reviewers and the set of reviewers that participated in the discussion, before and after the discussion, was small.”

7. Regarding to the two observations that the authors indicated on results section (see paragraphs 3 and 4 of page 8), any of them actually answer the research question of the paper. In my own opinion, to answer that question, authors should compare the initial and the final scores of reviewers within each experimental group not between groups. The conclusions that the authors point out are additional conclusions from the results.

Thank you for your suggestion, we updated the discussion section in the revised manuscript to reflect these observations, as mentioned in the previous answer. 

Further, we would like to note that in this work, we designed the experiment by having different initiators in the two groups specifically so that the difference in behavior between the two groups will allow us to isolate the effect of herding. Herding within a group, that is the change in initial and final scores within a group towards the initiator, does not necessarily indicate herding behavior. Herding to the initiator within a group could be because of many reasons other than herding. For instance, in our results, we see herding towards the mean in both groups. This implies that if the mean score on average is closer to that of the positive initiator, then we should conclude that there was herding towards the initiator in the group with the positive initiator (P+). However, that is not the case, since this herding behavior is explained by herding towards the mean which is seen in both the groups (P+ and P-). 

8. Finally, the discussion section should be extended. I think this section should indicate the main goal of the study, include a brief summary of the main results and conclusions, the theoretical and practical implications of the findings, the main limitations of the study, and some suggestions for future research.

Thank you for this suggestion. We have updated the Discussion section in the revised manuscript to reflect the points mentioned in this comment. We reproduce the updated text here: 

“The results obtained in our work show that \\revision{there is no evidence of herding in peer review. While our experiment managed to achieve an imbalance in the opinion of the discussion initiators across conditions, and despite past work having documented an undue influence of the first piece of information on the final decision [citations] in various other settings and applications, the difference in the acceptance rates is not significant and does not suggest the presence of herding behavior in peer review discussions. Regarding policy implications, the absence of the effect suggests that the absence of a unified approach towards discussion management does not result in an increased arbitrariness of the resulting decisions.” .. “These caveats present limitations of the study, and are found in the design of the intervention, the choice of papers for the experiment, the satisfaction of Requirement 2, the spurious correlations induced by reviewer identity, opinion of the discussion initiator and the possibility of alternative models for herding.”

Minor Comments:

9. I suggest clarifying what mean NS, AS and HD (see paragraph 9 of page 5), because it is not reported in the paper.

Thank you for pointing out this lack of clarity in this sentence. We were using these abbreviations to refer to the authors of this paper. We have edited the manuscript to make it clear. NS = Nihar Shah, AS = Aarti Singh, HD = Hal Daumé III. 

10. In the section 3.1, the authors wrote: “Finally, we note that most of the papers used in the experiment had some discussion and the length of the discussion is similar across conditions (Rows 5 and 7)”, but the authors did not explain how the length of the discussion was measure. This is important to understand what meaning the length of the discussion (time spent on the discussion, paper extension of the discussion, …).

Thank you for pointing out this lack of clarity about this point. We clarify this in the revised manuscript, text reproduced after comment 11. The length of the discussion for a paper is measured as the number of messages in a paper’s discussion thread.

11. I suggest explaining the row 6 of table 1 in the results section.

Thank you for pointing this out, we have added an explanation of Row 6 in our results section, we reproduce the text here: 

“Finally, we note that most of the papers used in the experiment had some discussion (Row 5). Specifically, we see that the mean number of participants in a paper's discussion and the length of a paper's discussion is similar across the two conditions, where we measure the length as the number of messages in a paper's discussion thread (Rows 6 and 7).”

References: 

[1] E Bareinboim, J Pearl, Controlling Selection Bias in Causal Inference. Proceedings of the Fifteenth International Conference on Artificial Intelligence and Statistics, PMLR 22:100-108, 2012.

[2] M Moerbeek, S van Schie, What are the statistical implications of treatment non-compliance in cluster randomized trials: A simulation study. Journal of Statistics in Medicine. Volume 38, Issue 26, 2019

[3] McCoy CE. Understanding the Intention-to-treat Principle in Randomized Controlled Trials. West J Emerg Med. 2017 Oct;18(6):1075-1078. doi: 10.5811/westjem.2017.8.35985. Epub 2017 Sep 18. PMID: 29085540; PMCID: PMC5654877.

---

## [Decision Letter · Decision Letter 1]

14 Feb 2023

PONE-D-22-17535R1A Large Scale Randomized Controlled Trial on Herding in Peer-Review DiscussionsPLOS ONE

Dear Dr. Charvi Rastogi,

Thank you for submitting your manuscript to PLOS ONE. After careful consideration, we feel that it has merit but does not fully meet PLOS ONE’s publication criteria as it currently stands. Therefore, we invite you to submit a revised version of the manuscript that addresses the points raised during the review process.

We have received two reports on your revised version. As you will see, there is no herding in this revision; Reviewer 1 recommends accepting the paper, while Reviewer 2 recommends rejection. I am very grateful to both reviewers since they did an excellent job detecting weaknesses and suggesting improvements to the paper.

After carefully reading and reviewing the manuscript myself, I consider that you addressed every point raised by reviewer 1 and some of the ones raised by reviewer 2.

The pending issues are explained in the report of Reviewer 2. Reviewer 2 discusses 5 points in this report, and I will detail my judgement on each of these points.

Point 1 deals with the methods section, I think the paper explains every critical step to be replicated. Regarding the length of the sections, I truly believe that the authors' preference must prevail (whenever addressing all the critical points). With all respect to the reviewers, I have often suffered the pain of finding a reviewer with different preferences regarding how to organize my paper, but this must be part of the freedom of authors to communicate their contribution.

Point 2. The point of the paper is not to check whether the mentioned bias operates in this study. The previous analysis was the one required according to the design, so there is no point in asking for the separate analysis requested by the referee. In addition, note that even though the statistical test would fail to reject the null hypothesis, ``the absence of evidence does not imply evidence of the absence'.' I'm afraid I have to disagree with the reviewer's reasoning regarding this point.

Point 3. I agree with the referee that authors must report, for completeness, the magnitude of the differences between groups.

Point 4. The reluctance of authors to dig deeper regarding the recency and primacy effect is legit. The underpinning of the herding effect is beyond the scope of this paper. However, this point could be included in the discussion as a potential issue to be analyzed by future research.

Point 5. In my opinion, all the crucial aspects of the paper are included in the discussion, extending the details is again an author's decision, and I have no complaints, but including my suggestion from point 4 would also fulfil one of the additional reviewer's recommendations.

In a nutshell, I think the authors have markedly improved the manuscript in the first revision. Reviewer 1 is completely satisfied with your improvements, and regarding the major concerns raised by Reviewer 2, after carefully reading the paper, I detailed which ones also concern me. Thus, consider this report a conditional acceptance once you address my comments in points 3 and 4. If you disagree, please report and explain carefully why you do not follow my advice.

We look forward to receiving your revised manuscript.

Kind regards,

Jaume Garcia-Segarra

Academic Editor

PLOS ONE

Journal Requirements:

Reviewers' comments:

Reviewer's Responses to Questions

**Comments to the Author**

1. If the authors have adequately addressed your comments raised in a previous round of review and you feel that this manuscript is now acceptable for publication, you may indicate that here to bypass the “Comments to the Author” section, enter your conflict of interest statement in the “Confidential to Editor” section, and submit your "Accept" recommendation.

Reviewer #1: All comments have been addressed

Reviewer #2: (No Response)

2. Is the manuscript technically sound, and do the data support the conclusions?

Reviewer #1: Yes

Reviewer #2: Yes

3. Has the statistical analysis been performed appropriately and rigorously? 

Reviewer #1: Yes

Reviewer #2: Yes

4. Have the authors made all data underlying the findings in their manuscript fully available?

Reviewer #1: No

Reviewer #2: No

5. Is the manuscript presented in an intelligible fashion and written in standard English?

Reviewer #1: Yes

Reviewer #2: Yes

6. Review Comments to the Author

Reviewer #1: The revised manuscript is much clearer on the differentiation between the cognitive biases (for example anchoring and primacy effect) and the herding behavior that can result from these biases. The additions to the introduction strengthens the necessity of the research question. I am also very happy about the extension of the discussion by including the limitations that arise from the design of the experiment (especially requirement 2).

Reviewer #2: Thank you for the opportunity of having a second review of the manuscript. Also, thank you to the authors for addressing some of my comments, but I still need to clarify some of the author´s answers. Below, authors can find my new concerns and comments.

1. Author´s answer to the reviewer´s comment 2: We did not understand this comment by the reviewer. Could the reviewer please elaborate what was imprecise and confusing? Regarding the reviewer’s suggestion of being “shorter”: we tried to provide the details to be fully rigorous, so we do not understand what the reviewer wishes to condense or eliminate. The experimental conditions and implications are described under the heading “Treatment scheme” and further discussed under the “Details of the experiment” heading right after that.

Reviewer answer: I am sorry whether I have not managed to express myself well in this comment. Now, I will try to be more concise.

I think that the method section should explain to the readers how the authors developed the experiment in order to any researcher are interested in replicated the study can do that. Hence, for this reason I think that the explanation should explain the method development step by step. Also, I think that this section should be shorter, getting to the point and eliminating the less relevant information or even adding this less important information in a supplementary material file. For instance, some information of the “details of the experiment” subsection could be sending to the supplementary material file. Figure 1 could also be sending to the supplementary material file. The method section takes up 4 pages, as much as the results and discussion section together, which are the sections more important of the paper, from my point of view.

2. Author´s answer to the reviewer´s comment 3 and 4: We combine the responses to comments 3 and 4 since they are aimed at the same issue.

In our analysis, we include the cases where the reviewers decided to ignore the discussion phase or where the reviewer asked to initiate the discussion did not initiate the discussion. The reason for our approach is avoiding selection bias, we explain this in more detail in the next paragraph. Literature in the field of causal inference has a lot of discussion on non-compliance and how ignoring non-complying data points (selecting only complying data points) can lead to biased estimates [1,2]. Our analysis approach is the same as the standard “Intention to treat” analysis commonly applied in causal inference literature [3].

While an analysis on data where the reviewers complied with the experiment design may increase the power of the test, it is very important to note that such an analysis may lead to selection bias and hence violate the false alarm guarantees of the statistical analysis. Specifically, it is possible that the reviewers who comply with the experiment are statistically significantly different in their behavior compared to the reviewers who do not comply with the experiment. If this is the case, then compliance will cause confounding in the analysis that you are suggesting. For instance, consider an example where senior reviewers are more likely to comply with our intervention by initiating the discussion. This could be because senior reviewers are more likely to attend to their reviewing responsibilities. In this example, if we observe herding behavior, it may be explained by the seniority of the discussion initiator if senior reviewers are more likely to be well-calibrated in their reviewing decisions. Hence, the analysis would show herding behavior even though the reason for the effect is not herding.

Finally, the review mentions that these cases (where the reviewer did not comply with the experiment) do not satisfy the requirements of our experiment, however, that is not the correct interpretation of the requirements posed in the manuscript. The requirements as stated in our work are described for the two groups of papers created in our randomized controlled trial, and not for individual papers. Further, we show that our group-level requirements are satisfied in Table 2.

Reviewer answer: Thank you for pointing me out the selection bias. I agree with the authors for the necessity to avoid this bias in order to guarantee the statistical analysis. However, authors are basing their argumentation in the assumption that if they do not include the cases where the reviewers decided to ignore the discussion phase or where the reviewer asked to initiate the discussion did not initiate the discussion, the analyses could be biased by the selection bias. I encourage to the authors to repeat the analyses without those cases in order to check if really this bias is operating in this study. Only knowing the results of this suggesting analyses, authors can support their argumentation.

Regarding to satisfying the requirements of the experiment, according to this author´s answer and the results of Table 2 of the manuscript, they show that the group-level requirements are indeed satisfied, hence I do not understand why the authors wrote in the manuscript (see paragraph four of page 6) that a priori they cannot guarantee that the requirements are indeed satisfied in the real experiment. This is an example of why say in the comment 2 of the first review, that the method section is a little confuse. In the paragraph 4 of page 6, authors specified that in order to support the experimental methodology, in Section 3 they empirically check for satisfaction of the stated requirements. I suggest to the authors reformulate this paragraph and directly indicate that the group-level requirements were satisfied in this experiment as the Table 1 and 2 shown. Also, I think that authors should explain the results of these Tables in this section of the manuscript, so the comment 5 of the first review would be addressed too, using the results section only to answer the research question of the experiment, that is the target of the study.

3. Author´s answer to the reviewer´s comment 6: We provide the difference between the initial and final mean scores of (i) initiators, (ii) reviewers and (iii) reviewers that participated in the discussion in Table 3. Further, we provide the initial mean scores of (i) initiators in Row 1 of Table 2, (ii) all reviewers in Row 2 of Table 1 and (iii) reviewers that participated in the discussion in Row 4 of Table 1. This information is provided in this order based on the narrative of the discussion of results. The final mean score for the three groups of reviewers can be straightforwardly deduced from this information.

Further, we provide the p value of the difference across groups in these changes (punctuations) in Rows 2,3 and 4 in Table 3. Our results show that while the difference is significant among the initiators (p value < 0.001), it is not significant for all reviewers or for reviewers that participated in the discussion.

Finally, following your suggestion, we also highlight this in the concluding section. We reproduce the text here:

“Further, we observe that within each condition, the change in the mean scores of all the reviewers and the set of reviewers that participated in the discussion, before and after the discussion, was small.”

Reviewer answer: I still think that the authors should report the initial and final mean scores of initiators, reviewers and reviewers that participated in the discussion in Table 3, because this Table is the table that answer the research question of the experiment. Moreover, although potential readers will be able to straightforwardly deduce this information from the data of Table 1 and 2, authors can facilitate the comprehension of the study if they directly report these data on Table 3. Also, I suggest to authors indicating the d value (Cohen, 1977) of the differences between groups because this is a statistic that indicates de magnitude of the differences and not only if indeed there are or there are not differences. Reporting the p value is important to know if indeed the differences are statistically significant or are due to random, but we need also know if these differences are small, medium, or large and what is the specific magnitude of the effect.

4. Author´s answer to the reviewer´s comment 7: Thank you for your suggestion, we updated the discussion section in the revised manuscript to reflect these observations, as mentioned in the previous answer.

Further, we would like to note that in this work, we designed the experiment by having different initiators in the two groups specifically so that the difference in behavior between the two groups will allow us to isolate the effect of herding. Herding within a group, that is the change in initial and final scores within a group towards the initiator, does not necessarily indicate herding behavior. Herding to the initiator within a group could be because of many reasons other than herding. For instance, in our results, we see herding towards the mean in both groups. This implies that if the mean score on average is closer to that of the positive initiator, then we should conclude that there was herding towards the initiator in the group with the positive initiator (P+). However, that is not the case, since this herding behavior is explained by herding towards the mean which is seen in both the groups (P+ and P-).

Reviewer Answer: According to the method section of the manuscript, I have understood that a set of papers was split into two experimental groups, one denominated P+ (where reviewer with the most positive initial opinion is asked to initiate de discussion) and another denominated P- (where reviewer with the most negative initial opinion is asked to initiate de discussion), hence any paper is simultaneously discussed in two experimental groups. Then, why the authors did not compare the initial and final mean scores of reviewers within each experimental group? I understand that comparing scores between experimental groups, we can estimate whether there are or there are not herding effect, but I think that comparing the scores within each experimental groups, authors can find interesting data regarding to recency and primacy effect. Whether the final mean score of the reviewers is statistically different to the initial mean score of the reviewers but is similar and not statistically different to the mean score of the first reviewer than contribute to the discussion with the most opposite opinion to the initiator opinion would be indicating a recency effect, in the opposite case would be indicating a primacy effect. I think that the authors should consider doing these analyses.

5. Author´s answer to the reviewer´s comment 8: Thank you for this suggestion. We have updated the Discussion section in the revised manuscript to reflect the points mentioned in this comment.

Reviewer Answer: Thank you to the authors for addressing changes in the manuscript regarding to this comment. However, I think that the authors still can address more changes in this sense. For instance, I miss some references to the main limitations of the study. Also, I miss some suggestions for future research. I also think that authors should extended the discussion about the main conclusions that they find and about the theorical and practical implications of the results. How does this paper contribute to the literature? Why this study is original?

References:

[1] Cohen J. Statistical power analysis for the behavioral sciences. New York (US): Lawrence Erlbaum; 1977. p. 567.

7. PLOS authors have the option to publish the peer review history of their article (what does this mean?). If published, this will include your full peer review and any attached files.

Reviewer #1: No

Reviewer #2: No

---

## [Author Response · Author response to Decision Letter 1]

26 Apr 2023

Authors' response: Thank you to the Academic Editor and the reviewers for their suggestions and positive feedback about our work! We have incorporated your suggestions in our manuscript. We provide a detailed response to your comments below. 

>> AE: In a nutshell, I think the authors have markedly improved the manuscript in the first revision. Reviewer 1 is completely satisfied with your improvements, and regarding the major concerns raised by Reviewer 2, after carefully reading the paper, I detailed which ones also concern me. Thus, consider this report a conditional acceptance once you address my comments in points 3 and 4. If you disagree, please report and explain carefully why you do not follow my advice.

>>Point 3. I agree with the referee that authors must report, for completeness, the magnitude of the differences between groups.

Authors' response: As suggested by reviewer 2, we have now provided the initial and final mean scores for each group of reviewers in the manuscript. The initial mean scores for different sets of reviewers had been provided in Table 1 (Row 2 and Row 4) and Table 2 (Row 1). In our revision, we provide the final mean scores for each set of reviewers in Section 3.3, while discussing Table 3. We reproduce the text here: 

“To understand the effect of the experiment, we first note the observed mean final scores in different sets of reviewers corresponding to groups P+ and P−. Among the reviewers who initiated the discussion, the mean final scores were 4.13 and 2.96 respectively. Among the reviewers who participated in the discussion, the mean final scores were 3.53 and 3.53, and among all reviewers, the mean final scores were 3.47 and 3.48 in P+ and P− respectively.” 

>>Reviewer 2 in point 3: “Also, I suggest to authors indicating the d value (Cohen, 1977) of the differences between groups because this is a statistic that indicates de magnitude of the differences and not only if indeed there are or there are not differences. Reporting the p value is important to know if indeed the differences are statistically significant or are due to random, but we need also know if these differences are small, medium, or large and what is the specific magnitude of the effect.”

Authors' response: Following this suggestion, we add the corresponding d value for each row in Table 3 to indicate the magnitude of the effect size. 

>>Point 4. The reluctance of authors to dig deeper regarding the recency and primacy effect is legit. The underpinning of the herding effect is beyond the scope of this paper. However, this point could be included in the discussion as a potential issue to be analyzed by future research.

Authors' response: To address this point in our manuscript, we added the following text to the discussion section: 

“In this work, we focused on the manifestation of herding effect in peer-review discussions. In future work, it is of interest to understand whether the separate cognitive biases attributed to herding behavior such as anchoring bias [35–39] and, primacy and recency effect [40] have any individual role to play in peer-review discussions.”

---

## [Editor Report · Decision Letter 2]

6 Jun 2023

A Large Scale Randomized Controlled Trial on Herding in Peer-Review Discussions

PONE-D-22-17535R2

Dear Dr. Rastogi,

We’re pleased to inform you that your manuscript has been judged scientifically suitable for publication and will be formally accepted for publication once it meets all outstanding technical requirements.

Kind regards,

Jaume Garcia-Segarra

Academic Editor

PLOS ONE
---

## [Editor Report · Acceptance letter]

19 Jun 2023

PONE-D-22-17535R2 

A Large Scale Randomized Controlled Trial on Herding in Peer-Review Discussions 

Dear Dr. Rastogi:

I'm pleased to inform you that your manuscript has been deemed suitable for publication in PLOS ONE. Congratulations! Your manuscript is now with our production department. 

Kind regards, 

on behalf of

Dr. Jaume Garcia-Segarra 

Academic Editor

PLOS ONE